# Macroscopic Spatial Analysis of the Impact of Socioeconomic, Land Use and Mobility Factors on the Frequency of Traffic Accidents in Bogotá

Alejandro Sandoval-Pineda [1,*] , Cesar Pedraza [2] and Aquiles E. Darghan [1]

1   Faculty of Agricultural Sciences, Universidad Nacional de Colombia, Bogotá 111321, Colombia
2   Department of Systems and Industrial Engineering, Universidad Nacional de Colombia, Bogotá 111321, Colombia
*   Correspondence: alsandoval@unal.edu.co

**Abstract:** The urban structure of a city, defined by its inhabitants, daily movements, and land use, has become an environmental factor of interest that is related to traffic accidents. Traditionally, macro modeling is usually implemented using spatial econometric methods; however, techniques such as support vector regression have proven to be efficient in identifying the relationships between factors at the zonal level and the frequency associated with these events when considering the autocorrelation between spatial units. As a result of this, the main objective of this study was to evaluate the impact of socioeconomical, land use, and mobility variables on the frequency of traffic accidents through the analysis of area data using spatial and vector support regression models. The spatial weighting matrix term was incorporated into the support vector regression models to compare the results against those that ignore it. The urban land of Bogotá, disaggregated into the territorial units of mobility analysis, was delimited as a study area. Two response variables were used: the traffic accidents index on the road perimeter and the traffic accidents index with deaths on the road perimeter, to analyze the total number of traffic accidents and the deaths caused by them. The results indicated that the rate of trips per person by taxi and motorcycle had the greatest impact on the increase in total accidents and deaths caused by them. Support vector regression models that incorporate the spatial structure allowed the modeling of the spatial dependency between spatial units with a better fit than the spatial regression models.

**Keywords:** support vector regression; regression spatial models; spatial autocorrelation; traffic accidents; macroscopic variable impacts



## 1. Introduction

Safety is one of the components that is constantly monitored in mobility within the urban environment. Explaining variations in vehicle accident levels per spatial unit using covariates at the zonal level is an area of active research in the context of road safety. This type of research is labeled as macro/aggregate level analysis [1] and usually has the purpose of identifying the relationships between socioeconomic, environmental, mobility, and land use factors and the frequency of traffic accidents observed in the spatial unit.

Different spatial units have been used in the macro analysis of traffic accidents, such as traffic analysis zones (TAZ) [2–11], census wards [12], census tracts [13–15], basic geo-statistical areas (BGEA) [16], etc., seeking to establish links between the covariates of the urban environment and the frequency of these events.

Some researchers even try to differentiate the macro processes that lead to the severity of traffic accidents. For example, [4] investigated the relationship between produced and attracted trips in different types of transport and the frequency of traffic accidents with serious injuries at the TAZ level in the state of Florida. In [12], the relationship between road infrastructure factors, socioeconomic and traffic characteristics, and traffic accidents

with minor, serious, and fatal injuries at the census ward level in London is analyzed. In [17], the importance of the variables associated with serious traffic accidents in four Florida counties is examined, and in [13], the relationship between the transportation infrastructure multimodal, socioeconomic, and land use variables and the frequency of fatal traffic accidents at the census track level is analyzed.

Traditional econometric methods are some of the most used in the macro analysis of traffic accidents. For example, negative binomial regression (NBR) models are very popular in this type of analysis as they adapt to the overdispersion of the frequency of events. In various studies, NBR models were developed to evaluate the relationship between sociodemographic variables, mobility, and road conditions, as well as the frequency of traffic accidents at the zonal level [2–4,6,8–10,13,16]. However, other authors differ from this method due to the inherent presence of spatial autocorrelation in traffic accidents that causes biases in the coefficients obtained. That is why they suggest the use of spatial econometric methods [12,18–20].

For their part, spatial econometric methods are based on the analysis of the dependence between observations in space and the use of a spatial weighting matrix W to represent the spatial arrangement of spatial units [21,22]. In the modeling of traffic accidents, there are different studies that use these analytic methods; for example, [15] uses the geographically weighted Poisson regression (GPWR) models for analysis, which are differentiated by the type of injury, using sociodemographic factors at the zonal level. Refs. [12,18–20] constructed models using spatial regression to identify the relationship between socioeconomic factors, road, and traffic characteristics and the frequency of vehicular accidents at the zonal level. They concluded that the use of spatial econometrical methods allows the explanation of the spatial autocorrelation in the data and eliminates the bias of the variables that present nonlinear relationships.

Another relatively new alternative to spatial econometric models for traffic accident analysis is support vector machines (SVMs). Machine learning methods are based on the idea of minimizing structural risk [23], and they have the great ability to address regression problems by identifying non-linear relationships between response and explanatory variables that cannot be captured by linear spatial regression models [11]. Different authors have used SVMs for the analysis of traffic accidents, obtaining better results with this technique, which has advantages over other traditional econometrics; for example, [24] compared the goodness of fit and the predictive performance of traffic accidents of SVM models with negative binomial models (NB), finding that the former predicted traffic accidents more effectively and accurately than the traditional NB models. On the other hand, [11] included spatial dependence in SVM models to assess the impact of socioeconomic and mobility factors on the probability of vehicular accident frequency. They found that spatial SVM models outperformed non-spatial ones, demonstrating the advantages of including spatial autocorrelation when modeling vehicle accident data.

In accordance with the previously described methods and covariates used for the macro analysis of traffic accidents, this research evaluated the impact of socioeconomic, land use, and mobility variables on the frequency caused by traffic accidents at the zonal level through spatial regression models and support vectors. The spatial autocorrelation term was included in the SVR models to assess the goodness of fit and the predictive performance compared to those that ignore it. The macro analysis was carried out using the territorial mobility analysis units (TMAUs) of the urban land of Bogotá, established by the district mobility secretariat (SDM), and the 2019 traffic accident records, differentiated by accidents as a spatial aggregation unit, including total vehicles and fatalities. The main findings of this research were: (1) the rate of trips per person by taxi and motorcycle had the greatest impact on the increase in total traffic accidents and deaths caused by them; (2) the support vector regression models that incorporate the spatial structure allowed the modeling of the spatial dependency between the spatial units with a better fit than the spatial regression models; (3) the variable constructed from the individual contributions of land uses and socioeconomical stratification (categorical variables) was relevant from

a statistical point of view and had a negative impact on the increase in deaths caused by traffic accidents; and (4) the variable rate of trips per person in the Transmilenio (BRT-type public transport) had an impact of reducing the total number of traffic accidents in the study area.

The rest of the document is structured as follows. The following section presents the study area and the data. Subsequently, part of the theory associated with the methods is described. Consecutively, the results and evaluation of the modeling are presented; the impacts of the covariates where the data provided evidence against null effects in their respective statistical hypotheses are discussed; and finally, the relevant conclusions and recommendations are presented.

## 2. Variables and Data

The response variables in the models were associated with the total number of traffic accidents and the deaths caused by them, based on the socioeconomic, land use, and mobility factors at the zonal level, using the territorial mobility analysis units (TMAUs) as a spatial unit. The study area was delimited to the urban land distributed in 110 TMAU polygons that corresponded to an approximate area of 37,972.7 Ha. The spatial information of the TMAU polygons and the alphanumeric information of the socioeconomic, mobility, and matrix indicators were extracted from the 2019 mobility survey of Bogotá and neighboring municipalities through the online services of the integrated information system on regional urban mobility (SIMUR). The georeferenced records of traffic accidents were extracted from the spatial database of Consolidated Road Accidents in Bogotá (Siniestros viales consolidados en Bogotá) of the Bogotá Open Data online services, filtering 29,028 records from the year 2019.

The integration of the socioeconomic, mobility, and matrix alphanumeric information with the TMAU spatial units was carried out in the ArcGISversion 10.7.1 software licensed by ESRI Colombia. The response variables corresponded to the traffic accidents index on the road perimeter (TAI) and the traffic accidents index with deaths on the road perimeter (TADI). The construction of these variables was accomplished through multiple geoprocesses in ArcMap 10.7.1., using the geographic information of the Integral Road Network of Bogotá (MVI), available in the online services of Open Data Bogotá, and was calculated as shown in **Equations (1)** and **(2),** respectively.

$$TAI = \frac{number\ of\ traffic\ accidents\ per\ TMAU}{Total\ length\ in\ KM\ of\ road\ sections\ per\ TMAU} \tag{1}$$

$$TADI = \frac{number\ of\ deaths\ caused\ by\ traffic\ accidents\ per\ TMAU}{Total\ length\ in\ KM\ of\ road\ sections\ per\ TMAU} \tag{2}$$

These indexes relate the number of traffic accidents and the deaths caused by them with the perimeter of the road network contained in each spatial unit, providing a measure that is closer to the reality of the phenomenon of vehicular accidents in an urban environment.

The spatial distribution of the TAI and the TADI can be seen in Figure 1.

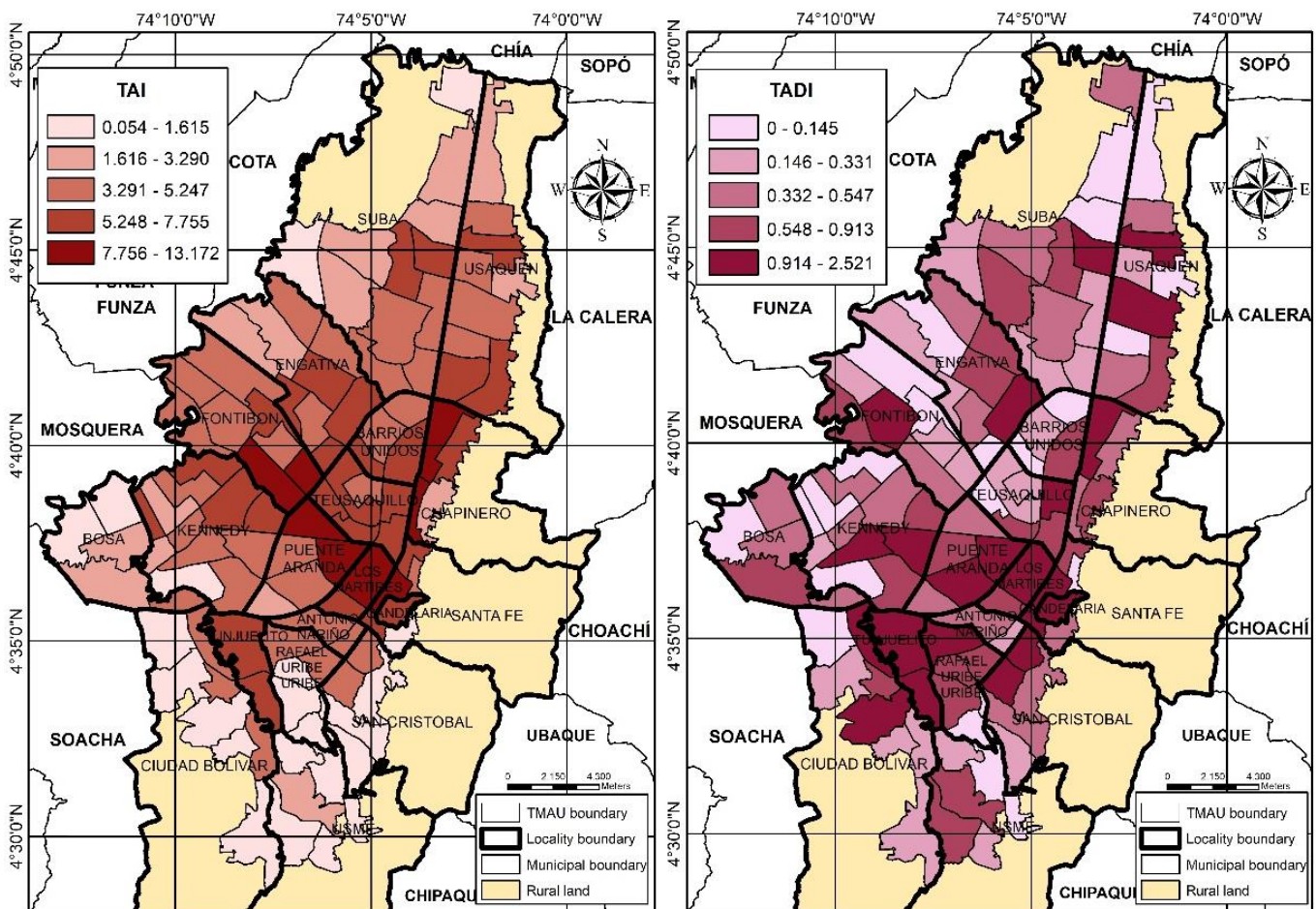

**Figure 1.** Spatial distribution of TAI (on the **left**) and TADI (on the **right**) in the study area. Authors' map.

## 2.1. Spatial Analysis Units and Data Aggregation

One of the determining interests of the present study was to determine the level of spatial aggregation of the data. As [12] points out, as the spatial unit of analysis is smaller, the counts per sampled unit decrease and the distribution becomes smaller. It becomes highly biased, which leads to a considerable increase in units with zero values [25].

For Bogotá, different spatial unit options were evaluated: localities, cadastral sectors, and TAZ, but the TMAUs, in addition to having the total coverage of urban land and being at an intermediate level between neighborhoods and localities, had information related to socioeconomic indicators, mobility, and land use; so, they were convenient for the analysis. In addition, these geographical divisions were established by the District Mobility Secretariat to analyze the city's mobility. In turn, these were made up of one or more neighborhoods that maintain homogeneous socioeconomic and land use conditions [26]. Table 1 shows the descriptive statistics of the variables involved.

**Table 1.** Descriptive statistics of the variables at the TMAU level.

| | Variables | Mean | Median | S.D. | Min | Max | C.V. |
|---|---|---|---|---|---|---|---|
| | **Response variables** | | | | | | |
| | Traffic accidents index on the road perimeter—TAI (traffic accidents/kilometer) | 4.303 | 4.369 | 2.635 | 0.055 | 13.172 | 0.612 |
| | Traffic accidents index with deaths on the road perimeter—TADI (traffic accidents with deaths/kilometer) | 0.584 | 0.422 | 0.521 | 0.000 | 2.521 | 0.893 |
| | **Land use factors** | | | | | | |
| LV | Land uses and socioeconomic stratification (weighted explained variance) | 0.55 | 0.44 | 0.60 | 0.11 | 4.07 | 1.09 |
| | **Socioeconomical factors** | | | | | | |
| X1 | Population density (people per kilometer) | 18,963.1 | 19,553.3 | 11,591.36 | 0 | 53,668.6 | 0.61 |
| X2 | Rate of motorization of motor vehicles—RMMV (motorized vehicles per 1000 inhabitants) | 236.6 | 212.46 | 139.66 | 0 | 753.43 | 0.59 |
| X14 | Number of households (households per TMAU) | 19,411.18 | 15,936.5 | 15,028.03 | 0 | 85,108 | 0.77 |
| | **Mobility factors** | | | | | | |
| X3 | Rate of pedestrian trips per person—RPTP (average daily pedestrian trips per person) | 2.13 | 2.14 | 0.36 | 0 | 2.79 | 0.17 |
| X4 | Rate of trips per person on public transport—RTPPT (average daily trips on public transport per person) | 0.59 | 0.6 | 0.18 | 0 | 1.05 | 0.31 |
| X5 | Rate of trips per person by taxi—RTPT (average daily taxi rides per person) | 0.1 | 0.08 | 0.07 | 0 | 0.3 | 0.70 |
| X6 | Rate of trips per person by car—RTPC (average daily car trips per person) | 0.33 | 0.27 | 0.29 | 0 | 1.5 | 0.85 |
| X7 | Rate of trips per person on motorcycle—RTPM (average daily motorcycle trips per person) | 0.08 | 0.08 | 0.04 | 0 | 0.25 | 0.55 |
| X8 | Rate of trips per person by bicycle—RTPB (average daily bicycle trips per person) | 0.1 | 0.08 | 0.07 | 0 | 0.33 | 0.68 |
| X9 | Trips in a typical day—origin—TTDO (origin of trips in a typical day across all available modes of transportation) | 114,176.69 | 100,169.93 | 75,521.33 | 551.97 | 394,626.84 | 0.66 |
| X10 | Trips in a typical day—destination—TTDD (destination of trips in a typical day across all available modes of transportation) | 114,241.15 | 101,562.74 | 75,297.79 | 551.97 | 395,196.43 | 0.66 |
| X11 | Travel rate per person in Transmilenio—RTPTM (average daily Transmilenio trips per person) | 4.48 | 1.56 | 11.08 | 0 | 102.86 | 2.47 |
| X15 | Average maximum speed allowed (kilometers per hour) | 39.55 | 39.16 | 6.06 | 30 | 54.61 | 0.15 |

SD: standard deviation; CV: coefficient of variation.

## 2.2. Socioeconomic Characteristics

According to the literature, different authors suggest that the frequency of traffic accidents is related to socioeconomic variables, such as population density, number of households, average family income, employed population, and households with and without a car [1,2,6–10,12,14,27]. In view of this, and to assess the impact of socioeconomic factors on traffic accidents, information was available on four socioeconomic variables that represent the exposed population, the number of vehicles per inhabitant, the number of households, and housing conditions (Table 1). These variables were obtained from the 2019 Mobility Survey of Bogotá and the neighboring municipalities for the year 2019 and reflect the socioeconomic profiles of the users and the preferences of citizens when traveling [26].

## 2.3. Mobility Characteristics

Mobility is a factor that is widely related to vehicular accidents as the movement of users is the factor that most likely causes these events to occur. Various authors have found that the production and attraction characterized by the reason for the trip and the type of transport is related to the frequency and severity of vehicular accidents in an urban environment [4,13,17,28]. In this research, the production and total attraction of the trips,

trips discriminated by the type of transport [26], and the average maximum speed allowed were involved. The latter was calculated at the zonal level from the vectorial information of the *Comprehensive Road Network of Bogotá* through multiple geoprocesses.

### 2.4. Land Use and Socioeconomic Stratum

The fact that traffic accidents are random events inherently related to the land use characteristics of the place where they occur is widely recognized. Multiple studies have shown the importance of land use in modeling traffic accidents [3,8,10,13,14,27,29]. This is related to the economic activities carried out daily by citizens that have substantial implications in the production and attraction of trips and therefore in the vehicular accident rate. In this research, information was available on land uses, including residential, commercial and services, and residential and industrial, and on socioeconomic strata: 1 (low-low), 2 (medium-low), 3 (medium-medium), 4 (medium), 5 (medium-high), and 6 (high) predominant in the TMAU.

## 3. Methods

The methods focused on the macro analysis of the relationships between the socioeconomic, land use, and mobility variables and the frequency of traffic accidents through the construction of spatial and support vector regression models. Spatial weighting matrices were introduced to reflect the spatial proximity relationships between the TMAU in the spatial regression and the spatial SVR models. A latent variable constructed from dimensions based on the individual contributions of the categorical variables of land use and socioeconomic strata was proposed. The selection of variables was carried out using the conventional backwards step-by-step method. The data were not partitioned into training and testing given that the interest of this study was to evaluate the effect of independent variables on the frequency of traffic accidents. The fit and performance of the models were evaluated using the mean absolute error (MAE) and the root mean square error (RSME) measures of performance (MOP) to compare spatial econometric models versus machine learning models in the macro analysis of traffic accidents. The effect of the relevant variables from a statistical point of view was analyzed by comparing the total impact of the spatial regression models and the weights derived from the linear kernel SVR models.

### 3.1. Spatial Autocorrelation

Spatial autocorrelation examines the degree to which a variable is correlated with itself at different locations [21]. The presence of this is usually detected by Moran's global I statistic that allows an analysis of the variations of spatial autocorrelation between neighboring values using the spatial weighting matrix W [30]. In the present research, the constructed W matrices were used from the rock contiguity criterion [31] and the relative neighbors' graph [32], using the mean centroid weighted by traffic accidents to model the spatial autocorrelation generated by the endogenous and exogenous variables and/or the error term. In addition, Moran's I test was used to assess the spatial independence in the model residuals. For this, the spatial dependence "*spdep*" package of R was used [33].

### 3.2. Traditional Linear Spatial Model

Spatial regression models study the effects of spatial interaction between geographic units. These effects can be generated by the endogenous and exogenous variables and/or the error term [22]. In this study, different spatial linear regression models were tested; however, the following two models obtained the best results to estimate the total number of traffic accidents and the deaths caused by them: the general nested spatial model (GNS) and the spatial Durbin error model (SDEM), respectively. These two models arise from a classical linear regression model (CLRM), where the residuals are $\varepsilon_i \sim \mathcal{N}\left(0, \sigma^2\right)$ and follow the traditional Gauss–Markov assumptions. Going from the general to the specific

approach, the general nested spatial (GNS) model includes all types of spatial interaction effects and is denoted as (**Equation (3)**).

$$Y = \rho WY + \alpha 1_N + X\beta + WX\theta + u; \ \ u = \lambda Wu + \epsilon \tag{3}$$

where $\rho$ is the autoregressive spatial coefficient, $\lambda$ is the spatial autocorrelation coefficient, $\theta$ and $\beta$ represent a $K \times 1$ vector of fixed but unknown parameters to be estimated. $W$ is a nonnegative $N \times N$ matrix that describes the spatial configuration or arrangement of the units in the sample and is called the spatial weighting matrix. Imposing the restriction of $\rho = 0$, the spatial error model (SDEM) is derived, which combines the exogenous interaction effects and the error term (**Equation (4)**).

$$Y = \alpha 1_N + X\beta + WX\theta + u \ ; \ u = \lambda Wu + \epsilon \tag{4}$$

For the construction of the spatial regression models, the spatial regression analysis package "*spatialreg*" of R [34] was used.

Total Impacts

Interpreting the impact or effect of the changes in an endogenous or exogenous variable in spatial regression models is complex as it differs in all regions or observations; however, [35] found how to summarize the impacts in all the regions with scalar measures called direct, indirect, and total impacts. The direct impacts summarize the sum of the impacts in the region experiencing a change; the indirect ones record the sum of the impacts due to changes in other regions; and the totals are the sum of the first two and summarize the impact of a region versus its change and that observed due to the change in other regions. In the present investigation, the total impact (**Equation (5)**) of the exogenous variables in the total number of traffic accidents and the deaths caused by them were calculated.

$$\vec{M}(r)_{total} = n^{-1} l'_n S_r(W)_{l_n} \tag{5}$$

where $S_r(W)$ acts as a "multiplier" matrix applying higher order neighborhood relations to $X_r$ y $l_n$ is a vector of ones of $n \times 1$. The total impacts derived from the spatial regression models were calculated with the spatial dependence "*spdep*" package of R [33].

*3.3. Support Vector Regression Models*

Support vector regression (SVR) is an efficient supervised learning tool for estimating real valued functions. Specifically, the SVR is formulated as an optimization problem (**Equation (6)**) in which a convex loss function to be minimized is defined. The idea is to find the flattest tube or margin that contains most of the training instances [23]. The hyperplane is represented in terms of support vectors that correspond to training samples that lie outside the boundary of the tube.

$$mn\frac{1}{2}\|w\|^2 + C\sum_{i=1}^{n}(\xi_i + \xi_i^*) \tag{6}$$

where $w$ is the magnitude of the vector or hyperplane, $C > 0$ (cost) determines the balance between the regularity of f and the amount up to which the deviations greater than $\varepsilon$ are tolerated, and $\varepsilon$ and $\xi_i$ and $\xi_i^*$ are the variables that control the error made by the regression function by approximating the margin. Thus, the approximation function $f(x)$ is described in (**Equation (7)**).

$$f(x) = \sum_{i=1}^{n}(\alpha - \alpha^*)K(x_i, \ x) \tag{7}$$

where $\alpha, \alpha^*$ are the dual variables associated with the constraints, with $\alpha, \alpha^* \ \epsilon \ [0, C]$, $i = 1, \ \dots, \ n, \ \sum_{i=1}^{n}(\alpha_i + \alpha_i^*) = 0 \ a\}$, and $K(x_i, x)$ is the kernel function. Two types of SVR

models were created in the present study: non-spatial and spatial. In the case of the spatial SVR models, the matrix $W$ was incorporated as a known input parameter lagging the independent variables with the criteria mentioned in Section 3.1., with the function *lag.listw* from the spatial dependence "*spdep*" package of R [33]. Two (2) kernel functions were used: the linear (**Equation (8)**) and the Gaussian radial basis (**Equation (9)**).

$$K(x, x_i) = x^T x_i \tag{8}$$

$$K(x, x_i) = \exp -\frac{\|x - x_i\|^2}{\sigma^2} \tag{9}$$

The first is to interpret the coefficients (weights) as impacts as this function does not extend the feature space; so, the resulting hyperplane is in the same input space. The second is to explore the predictive performance of the models. For the construction of the SVR models, the linear predictive models package "*LiblineaR*" of R [36] was used.

## 4. Analysis of Results

The relationship between socioeconomic, land use, and mobility factors and the total number of traffic accidents and deaths caused by these, represented by the traffic accidents index on the road perimeter (TAI) and the traffic accidents index with deaths on the road perimeter (TADI), was analyzed by means of spatial econometric models (SDEM and GNS) and SVR models (non-spatial and spatial). The spatial weighting matrices mentioned in Section 3.1 were used to model each response variable. The selection of variables in the spatial regression models was carried out using the backward step-by-step staggering method. Finally, the SVR models were made up of the variables that were statistically relevant in the spatial regression models.

### 4.1. Latent Variable Construction

In this research, an integration of the categorical variables was proposed: land uses and socioeconomic strata in a latent variable (LV) constructed from the individual contributions of each one of them through a multiple correspondence analysis (MCA) (**Equation (10)**). The sum of these contributions is usually related to the association patterns [37] that can be used as input in the modeling. This was conducted to quantify the effect of the underlying structure of these two categorical variables on the total number of accidents and deaths caused by them.

$$LV = \frac{1}{c} \sqrt{\sum_{i=1}^{d} (D_i f_i)^2} \tag{10}$$

where $c$ is a constant that represents the total percentage of the explained variance from extracted dimensions, $D_i$ represents the $i$-th variance explained by dimension, and $f_i$ is the variance explained by the $i$-th dimension normalized with the total variance explained by the extracted dimensions. For the current case, an explained variance of 81.2% was obtained from six dimensions (**Equation (11)**), which is above the recommended threshold of 70% [38].

$$LV = \sqrt{\left(Dim1 \times \tfrac{15.9}{81.2}\right)^2 + \left(Dim2 \times \tfrac{15.3}{81.2}\right)^2 + \left(Dim3 \times \tfrac{14.2}{81.2}\right)^2 + \left(Dim4 \times \tfrac{12.5}{81.2}\right)^2 + \left(Dim5 \times \tfrac{12.5}{81.2}\right)^2 + \left(Dim6 \times \tfrac{10.8}{81.2}\right)^2} \tag{11}$$

### 4.2. Spatial Regression Models

The results of the models to estimate the variables TAI and TADI, together with the coefficient of determination, the heteroscedasticity test (Breusch–Pagan), normality of residuals (Shapiro–Wilk), and the spatial autocorrelation (Monte Carlo simulation Moran's I), and the MOP are presented in Table 2. The dependence between variables was reviewed, identifying the non-presence of perfect multicollinearity in the models.

**Table 2.** Results of the spatial regression models for the variable TAI.

| | | TAI~ | | | TADI~ | | |
| | | GNS | | | SDEM | | |
| | Variables | Coeff | * | E.E. | Coeff | * | E.E. |
|---|---|---|---|---|---|---|---|
| | Intercept | $-5.0505$ | *** | 1.3377 | 0.0521 | — | 0.2389 |
| | **Land use factors** | | | | | | |
| LV | Land uses and socioeconomic stratification | — | — | — | $-0.1750$ | * | 0.0848 |
| W(LV) | Land uses and socioeconomic stratification | — | — | — | $-0.3116$ | * | 0.1320 |
| | **Socioeconomic factors** | | | | | | |
| X14 | Number of households | $-6.77 \times 10^{-5}$ | *** | $1.61 \times 10^{-5}$ | — | — | — |
| W(X1) | Population density | $-6.23 \times 10^{-5}$ | * | $2.76 \times 10^{-5}$ | — | — | — |
| W(X2) | RMMV | — | — | — | 0.0034 | * | 0.0015 |
| W(X14) | Number of households | — | — | — | $-1.40 \times 10^{-5}$ | * | $6.94 \times 10^{-6}$ |
| | **Mobility factors** | | | | | | |
| X5 | RTPT | 8.0470 | * | 3.5763 | 1.7509 | * | 0.7265 |
| X6 | RTPC | $-1.4546$ | * | 0.5979 | -0.5181 | ** | 0.2009 |
| X8 | RTPB | — | — | — | — | — | — |
| X9 | TTDO | $1.47 \times 10^{-5}$ | *** | $2.99 \times 10^{-6}$ | $-3.12 \times 10^{-5}$ | * | $1.47 \times 10^{-5}$ |
| X10 | TTDD | — | — | — | $3.35 \times 10^{-5}$ | * | $1.48 \times 10^{-5}$ |
| X11 | RTPTM | $-0.0315$ | * | 0.0146 | — | — | — |
| X15 | Average maximum allowable speed | 0.1585 | *** | 0.0286 | — | — | — |
| W(X5) | RTPT | 10.4450 | . | 6.1049 | — | — | — |
| W(X6) | RTPC | — | — | — | $-1.6902$ | * | 0.7569 |
| W(X7) | RTPM | 11.4270 | . | 7.4649 | 2.7204 | . | 1.4877 |
| W(X8) | RTPB | 10.9880 | *** | 4.1595 | — | — | — |
| W(X10) | TTDD | — | — | — | $3.00 \times 10^{-6}$ | * | $1.17 \times 10^{-6}$ |
| W(X11) | RTPTM | $-0.0932$ | *** | 0.0274 | — | — | — |
| | $R^2$ | 0.7263 | | | 0.3571 | | |
| | $\rho$ | 0.2607 | | | — | | |
| | $\lambda$ | $-0.2930$ | | | 0.2222 | | |
| | Log Likelihood | $-190.8752$ | | | $-59.6248$ | | |
| | Moran I (Residuals) | 0.5050 | | | 0.4060 | | |
| | Shapiro–Wilk (Residuals) | 0.5488 | | | 0.0895 | | |
| | Breusch–Pagan | 0.0503 | | | 0.0600 | | |
| | MAE | 1.0711 | | | 0.2762 | | |
| | RSME | 1.3500 | | | 0.3690 | | |

W: spatially lagged variable; statistically significant at 0.0001 '***', 0.001 '**', 0.01 '*', 0.05 '.'; '—'not included in the model.

Regarding the TAI, in the GNS model all the assumptions of the MRCL model were met; the parameter $\rho$ indicates that 26.07% of the alterations in the traffic accidents index on the road perimeter in a TMAU affect the probability of accidents in the neighboring TMAUs. Moreover, the parameter $\lambda$ shows that as the random error increases in a TMAU, it decreases in the adjacent TMAUs at a ratio of 29.3%. In addition, 72.63% of the TAI can be explained by population density, the number of households, the rate of trips per person by taxi, car, motorcycle, bicycle, and Transmilenio, trips on a typical day of origin, and the average number of trips per person at the maximum permitted speed.

Regarding the TADI, the SDEM model complied with all the assumptions of the MRCL model. This had values of 0.2762 and 0.3690 in the MOP MAE and RMSE, respectively. The parameter $\lambda$ indicates that as the random error increases in a TMAU, it increases in the adjacent TMAUs at a ratio of 22.22%. From this model, it can be interpreted that

35.71% of the TADI can be explained by land use and socioeconomic stratification, the rate of motorized vehicles, the number of households, the rate of trips per person by taxi, automobile, and motorcycle, and travel during a typical day from origin and destination.

### 4.3. Support Vector Regression Models

Starting from the variables that were relevant from a statistical point of view in the spatial regression models, the results of the non-spatial and spatial SVR models with the variables that were selected from the spatial regression models, their effects (weights), the optimized hyperparameters (C, L1–L2, $\sigma$ and $\epsilon$), and the MOPs are presented in Table 3.

**Table 3.** Results of the SVR models with linear kernel and radial basis for the variables TAI and TADI.

| | Variables | TAI~ | | | | TADI~ | | | |
|---|---|---|---|---|---|---|---|---|---|
| | | Linear | | Radial Basis | | Linear | | Radial Basis | |
| | | SVR NE | SVR E | SVR NE | SVR E | SVR NE | SVR E | SVR NE | SVR E |
| | Bias **(b)** | 4.2983 | 4.2686 | — | — | 0.5818 | 0.5858 | — | — |
| | **Land use factors** | | | | | | | | |
| LV | Land uses and stratification socioeconomic | — | — | — | — | −0.0697 | −0.0840 * | × | × |
| | **Socioeconomic factors** | | | | | | | | |
| X1 | Population density | — | 0.4568 * | — | × | — | — | — | — |
| X2 | RMMV | | | | | — | 0.1249 * | — | × |
| X14 | Number of households | −1.2096 | −0.9718 * | × | × | — | −0.0640 * | — | × |
| | **Mobility factors** | | | | | | | | |
| X5 | RTPT | 1.0579 | 1.2382 * | × | × | 0.1030 | 0.0364 * | × | × |
| X6 | RTPC | −0.3987 | -0.3610 * | × | × | −0.0998 | −0.1063 * | × | × |
| X7 | RTPM | −0.0627 | 0.1221 * | × | × | — | 0.0587 * | | × |
| X8 | RTPB | 0.3901 | 0.5337 * | × | × | — | — | — | — |
| X9 | TTDO | 1.3135 | 1.1058 * | × | × | 0.0442 | 0.1781 * | × | × |
| X10 | TTDD | — | — | — | — | 0.1079 | 0.0870 * | × | × |
| X11 | RTPTM | −0.2491 | −0.5991 * | × | × | — | — | — | — |
| X15 | Average maximum speed allowed | 1.0128 | 0.8714 * | × | × | — | — | — | — |
| | **Hyperparameters** | | | | | | | | |
| | C (cost) | 1 | 0.4444 | 1 | 1 | 0.1111 | 0.1111 | 0.2222 | 1 |
| | L1–L2 (Loss function) | L2 | L2 | — | — | L2 | L2 | — | — |
| | $\sigma$ (sigma) | — | — | 0.5000 | 0.5000 | — | — | 0.5000 | 0.5000 |
| | $\epsilon$ (épsilon) | 0 | 0 | 0.1000 | 0.1000 | 0 | 0 | 0.1000 | 0.1000 |
| | $R^2$ | 0.5070 | 0.5128 | 0.7651 | 0.8252 | 0.1954 | 0.1995 | 0.2006 | 0.6908 |
| | Moran I (Residuals) | 0.0420 | 0.3220 | 0.0070 | 0.0540 | 0.1130 | 0.0520 | 0.1370 | 0.2990 |
| | MAE | 1.1462 | 1.2071 | 0.5572 | 0.4843 | 0.3629 | 0.3537 | 0.3117 | 0.1508 |
| | RMSE | 1.5023 | 1.4809 | 0.9637 | 0.8792 | 0.4655 | 0.4643 | 0.4640 | 0.2885 |

\* Spatially lagged variable, '—' not included in the model, '×' without assigned weight.

The results shown in Table 3 indicate that the non-spatial SVR model with the linear kernel and the optimal hyperparameters $C = 1$ and the loss function L2 explained approximately 50.7% of the TAI. Likewise, the non-spatial model with a radial base kernel and these same variables explained approximately 76.51% of the TAI. On the other hand, the spatial model with a linear kernel and the optimal hyperparameters $C = 0.4444$ and a loss function L2, made up of the variables RTPC (X6), TTDO (X9), and the spatially lagged variables of population density (X1), number of households (X14), RTPT (X5), RTPM (X7), RTPB (X8), TTDO (X9), RTPTM (X11), and average maximum speed allowed (X15), explained 51.28% of the TAI. In contrast, the spatial model with a radial base kernel and these same variables explained approximately 82.52% of the TAI. Of these models, only the spatial models removed the spatial dependence on the residuals with the variables specified in the models.

Regarding the TADI, the results indicated that the non-spatial SVR model with the linear kernel and the optimal hyperparameters $C = 0.1111$ and the loss function L2 explained approximately 19.54% of the TADI, while the spatial model with the radial base

kernel made up of these variables allowed the explanation of 20.06% of the TADI. On the other hand, the spatial SVR model with the linear kernel and the optimal hyperparameters $C = 0.1111$ and the loss function L2, made up of the RTPT (X5), the TTDO (X9), and the spatially lagged variables land use and socioeconomic stratification (LV), RMMV (X2), number of households (X14), RTPC (X6), RTPM (X7), and TTDD (X10), explained the TADI at approximately 19.95%. Finally, the model with the radial base kernel and these variables explained 69.08% of the TADI.

### 4.4. Impact Analysis

The impacts of the variables that were statistically relevant in the spatial regression models and that later formed the SVR models were analyzed by comparing the total impact of the spatial regression models and the weights derived from the SVR models with the linear kernel. The impact of the exogenous variables on the TAI is presented in Table 4.

**Table 4.** Impacts of exogenous variables on the TAI.

| | Variables | Impacts | | |
|---|---|---|---|---|
| | | GNS | SVR NE | SVR E |
| | **Socioeconomic factors** | | | |
| X1 | Population density | $-8.43 \times 10^{-5}$ | — | 0.4568 |
| X14 | Number of households | $-9.16 \times 10^{-5}$ | $-1.2096$ | $-0.9718$ |
| | **Mobility factors** | | | |
| X5 | Rate of trips per person by taxi (RTPT) | 25.0122 | 1.0579 | 1.2382 |
| X6 | Rate of trips per person by automobile (RTPC) | $-1.9674$ | $-0.3987$ | $-0.3610$ |
| X7 | Rate of trips per person by motorbike (RTPT) | 15.4555 | $-0.0627$ | 0.1221 |
| X8 | Rate of trips per person by bicycle (RTPB) | 14.8614 | 0.3901 | 0.5337 |
| X9 | Trips in a typical day—origin (TTDO) | $1.99 \times 10^{-5}$ | 1.3135 | 1.1058 |
| X11 | Ratio of trips per person in Transmilenio (RTPTM) | $-0.1688$ | $-0.2491$ | $-0.5991$ |
| X15 | Average maximum speed permitted | 0.2144 | 1.0128 | 0.8714 |

'—' Impact not calculated.

Regarding the TAI, most of the variables were found to be relevant from the statistical point of view and had the expected sign, which was consistent with the other studies [2,4,5,8,10,12,17,39]. Specifically, RTPT was the variable with the greatest impact on the increase in TAI and coincided with what was found in [17], in which it was found that this was one of the variables with the greatest importance in the increase in total accidents in the state of Florida. The RTPM had a positive impact on the increase in the TAI, a finding that is in line with [40], in which it was found that the risk of accidents and the severity of injuries associated with the use of motorcycles was significantly higher than that associated with any other type of vehicle. Similarly, and in line with what was found by [39], the RTPB had a direct association with total traffic accidents. These data are consistent with the fact that this type of transport is associated with being one of the most vulnerable (along with pedestrians).

Moreover, the average maximum speed allowed had a positive impact on the increase in TAI, as has been found in multiple studies [1,7,8,41]. A higher speed leads to a substantial increase in the risk of vehicular accidents; so, the impact found was consistent with the literature. Similarly, the TTDOs had an impact on the increase in the traffic accidents index on the road perimeter, which is something that coincides with the finding of [28], which supports this as most road trips are carried out in private vehicles during peak hours and on a routine basis and, therefore, have a greater exposure to risk. Likewise, as the results indicate, the effect of population density is neutral (very close to zero) and positive in the increase in the TAI in the spatial and vector support regression models, respectively, which is expected because in studies such as [2,6] it has been found that population density is

associated with an increase in traffic accidents as a larger population is always consistent with a greater opportunity in terms of exposure and risk of vehicular accident.

In contrast, an unexpected result in the traffic accidents index on the road perimeter models was that it was the RTPC that had the greatest impact on the reduction in this (a negative association), which suggests that the number of accidents per KM decreases with the increase in RTPC at the TMAU level. This is not consistent with the hypothesis that traffic accidents increase with an increase in light car trips and does not coincide with the findings in [4,17,42]. However, a possible explanation for this is that, even though in the decade between 2007 and 2017 the number of light vehicles rose by approximately 51%, during the last four years, traffic accidents with light vehicles involved have decreased by 38.81% [43], a downward trend. In the same way, the number of households had a negative impact on the increase in TAI, which was unexpected as studies such as [8,10] identified that the increase in this variable is related to a larger population, which produces a greater number of trips and a greater probability of vehicular accidents.

The RTPTM was one of the variables of interest in this study as the Transmilenio is a type of transport called Bus Rapid Transit (BRT) that usually travels through exclusive lanes and has other characteristics that differentiate it from the others. The impact of reducing the traffic accidents index on the road perimeter of the RTPTM was consistent with [44], in which it was identified that the frequency of accidents decreases as BRT trips increase. One reason for this may be because BRTs allow the transportation of a greater number of people, reducing traffic volume and the risk of vehicular accidents. The foregoing makes sense as 2,058,888 Transmilenio trips are made daily, representing 15.38% of the total trips in Bogotá [43]. The impact of the exogenous variables on the TADI is presented in Table 5.

**Table 5.** Impacts of exogenous variables on the TADI.

| | Variables | Impacts | | |
|---|---|---|---|---|
| | | SDEM | SVR NE | SVR E |
| LV | **Land use factors** Land uses and socioeconomic stratification | −0.4866 | −0.0697 | −0.0840 |
| | **Socioeconomic factors** | | | |
| X2 | Rate of motorization of motor vehicles (RMMV) | 0.0034 | — | 0.1249 |
| X14 | Number of households | $-1.40 \times 10^{-5}$ | — | −0.0640 |
| | **Mobility factors** | | | |
| X5 | Travel tax per person by taxi (RTPT) | 1.7509 | 0.1030 | 0.0364 |
| X6 | Travel rate per person by car (RTPC) | −2.2083 | −0.0998 | −0.1063 |
| X7 | Travel rate per person on motorcycle (RTPM) | 2.7204 | — | 0.0587 |
| X9 | Typical day trips—origin (TTDO) | $-3.12 \times 10^{-5}$ | 0.0442 | 0.1781 |
| X10 | Typical day trips—destination (TTDD) | $3.65 \times 10^{-5}$ | 0.1079 | 0.0870 |

'—' Impact not calculated.

Most of the variables that were relevant in the TADI had the expected sign. As with the TADI models, one of the variables with the greatest impact on the increase in deaths due to traffic accidents was the RTPM, which is consistent with [40], in which it was found that the severity of injuries in vehicle accidents associated with the use of motorcycles was significantly greater than that related to other vehicles. Similarly, and as expected, the RTPT had a positive impact on the increase in deaths due to traffic accidents, a fact that is partially consistent with [17] in which it was found that trips generated and attracted by taxis are important in serious vehicle accidents.

Similarly, the RMMV was another variable that was expected to have an impact on the increase in TADI. This was consistent with the assumption that deaths from traffic accidents increase with the increase in motorized vehicles per inhabitant; in addition, this can be explained by the fact that in the decade between 2007 and 2017 the vehicular fleet in Bogotá increased by 54.1% [43], and the average number of deaths due to vehicle accidents was 553.72 deaths per year, a figure that, although small, exceeds the number of deaths

due to traffic accidents by 0.49% compared to the traffic of 2007. Likewise, the TTDO and TTDD had a positive impact on the increase in deaths due to traffic accidents. In [28], it is indicated the production and daily attraction of trips are generally caused by work, education, and shopping, which is why they are routinely carried out in peak periods, in a greater hurry, and sometimes over greater distances, generating greater exposure to the risk of being involved in a severe vehicular accident.

The RTPC was the variable with the greatest impact on the reduction in the traffic accidents index with deaths on the road perimeter. This result is consistent with the trend of the period between 2015 and 2019, in which the number of deaths in light vehicles involved in traffic accidents was reduced by 38% [45]. This can be explained in part by the fact that in 2017 it was mandatory to improve active and passive safety measures in light vehicles regarding the integration of the ABS (anti-lock) braking system, the air bag (airbag), and head restraints (Resolution No. 3752, 2015). The variable of the number of households had an unexpected result, because, as [8,10] point out, an increase in the number of households has repercussions for a larger population and therefore a greater number of users that are vulnerable to suffering an accident in traffic that can be deadly.

Finally, an interesting finding was that the variable constructed from the individual contributions of land uses and socioeconomic stratification was relevant from a statistical point of view as it was able to discriminate the classes represented in a numerical value that shows association patterns related to these categorical variables. The negative impact of this on the traffic accidents index with deaths on the road perimeter can be explained by various causes: (1) 79.04% of the urban land in Bogotá is predominantly residential and, as [3] points out, the use of residential land is related to traffic accidents with minor injuries and (2) 73.71% of the urban land of Bogotá has as predominant strata one, two and three, strata where the use of public transport predominates, which only had 22% of the total traffic accident fatalities in the city in 2019 [45].

The findings of this study made it possible to identify the factors with the greatest impact on the increase in traffic accidents and the deaths caused by them in Bogotá. This can serve the entities in charge of road safety in the city to focus their attention on factors such as the active and passive safety of taxi transport, the high level of vulnerability of motorcycle and bicycle transport users, the control of the maximum speed allowed, and the incentive of the use of mass public transport of Transmilenio to meet the objective of reducing the number of victims due to traffic accidents as much as possible.

## 5. Conclusions and Recommendations

This research evaluated the impact of socioeconomic, land use, and mobility variables on the frequency of traffic accidents at the TMAU level using spatial and support vector regression models. Moran's I contrast allowed the identification of the fact that the traffic accident data were spatially autocorrelated, confirming what was found in other studies and supporting the use of spatial regression models for the present analysis. These provided the use of analytic spatial methodologies to consider these dependency structures.

The variables that had a statistically significant relationship with the total number of traffic accidents in the study area were the following: population density, number of households, RTPT, RTPC, RTPM, RTPB, RTPTM, TTDO, and the average maximum speed allowed. These had a spatial relationship with the total number of traffic accidents represented by the contiguity of order one in the directions of the cardinal points (rock criterion). This allowed it to be shown that the population density, the number of households, the RTPT, and the RTPC have a greater effect in the neighboring TMAU than in the TMAU where there is a change in the traffic accidents index on the road perimeter, while the RTPM, RTPB, RTPTM, the TTDO, and the average maximum speed allowed had a greater effect in the TMAUs where there is a variation in the total number of traffic accidents than in those of their neighbors. Land use and socioeconomic stratification, the RMMV, the number of households, the RTPT, RTPC, RTPM, and the TTDO and TTDD had a relevant relationship with mortality due to traffic accidents. Moreover, they had a spatial relationship with the

mortality of traffic accidents, as represented by the graph of relative neighbors, and had a greater effect in the TMAU where there was a variation in the change in the traffic accidents index with deaths on the road than in the neighboring TMAUs.

The GNS model allowed us to consider the spatial autocorrelation present in the traffic accidents index on the road perimeter, the explanatory variables, and the error term with better results than the other traditional econometric models. The SDEM model made it possible to explain the traffic accidents index with deaths on the road perimeter by modeling the spatial autocorrelation present in the exogenous variables and the error term, complying with all the assumptions of the CLRM. On the other hand, the spatial SVR models with linear and radial basis kernel function allowed the elimination of the spatial dependence on the residuals. The spatial SVR model with a radial basis function obtained a better fit than the spatial regression and linear kernel SVR models.

The impact analysis allowed us to identify that the rate of trips per person by taxi, motorcycle, and bicycle were the variables with the greatest impact on the increase in total traffic accidents in the urban area of Bogotá, while the RTPC had the greatest impact on the reduction in these in the study area. Although this finding is not consistent with the literature, it can be explained by the trend in recent years of an inverse relationship between the growth of the vehicle fleet of light automobiles and the vehicle accidents in which they are involved. The RTPTM had a negative relationship with the increase in total traffic accidents, showing that the use of Transmilenio mass transportation reduces the frequency of vehicle accidents in the study area.

On the other hand, the rate of trips per person by taxi and motorcycle were the variables with the greatest impact on the increase in deaths due to traffic accidents in the urban area of Bogotá. The motorcycle was one of the main modes of transport historically that contributes more deaths in traffic accidents. On the other hand, the RTPC is the variable with the greatest negative relationship in the increase in deaths in traffic accidents, which reflects the fact that the mandatory regulation in active and passive safety measures for light vehicles considerably improved the mortality figures in vehicle accidents. Land use and socioeconomic stratification had an indirect negative relationship on the increase in deaths in traffic accidents.

Compared to traditional econometric methods, the proposed SVR provided a new perspective for regional-level traffic accident analysis, incorporating spatial proximity effects and providing results superior to those of spatial regression models. However, these results were obtained at the local level; evaluating this type of models with other datasets is recommended to validate the results obtained in future research. Likewise, because in the present study land uses were superficially analyzed due to their predominance in the TMAU, analysis of the individual impact of land uses on more detailed units, such as ZAT polygons, is recommended.

**Author Contributions:** Conceptualization, A.S.-P.; methodology, A.S.-P. and A.E.D.; software, A.S.-P.; validation, A.S.-P.; formal analysis, A.S.-P. and A.E.D.; investigation, A.S.-P., C.P. and A.E.D.; resources, A.S.-P.; data curation, A.S.-P.; writing—original draft preparation, A.S.-P.; writing—review and editing, A.S.-P.; visualization, A.S.-P.; supervision, C.P. and A.E.D.; project administration, A.S.-P. and C.P. All authors have read and agreed to the published version of the manuscript.

**Funding:** This research received no external funding.

**Institutional Review Board Statement:** Not applicable.

**Informed Consent Statement:** Not applicable.

**Data Availability Statement:** Data are contained within the article.

**Conflicts of Interest:** The authors declare no conflict of interest.

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
