# Peer review of "Macroscopic Spatial Analysis of the Impact of Socioeconomic, Land Use and Mobility Factors on the Frequency of Traffic Accidents in Bogotá"

_computers, doi:10.3390/computers11120180_

Round 1
Reviewer 1 Report
1. Equations (2-1) and (2-2) refers to UTAM. Is UTAM different than TMAU? If they are the same, please be consistent - use only either UTAM or TMAU, not both. If they are different, then, please define UTAM.
2. Please explain all variables in Table 2.1. For example, what constitute mean of Land Use Factors (LV)? When mean LV = 0.55, how is it computed, i.e. mean of what? What is the unit of LV?
3. How is W (spatial weighting matrix) computed? What is the equation for computing W? What does W represent? Is there any learning algorithm used to compute W?
4. Why GNS and SDEM selected as your spatial regression models? What are your justifications? Are they better than other spatial regression models?
5. In Line 225, the CLRM model requires the residual e~N(0,sd2). Nowhere in the paper you test whether the residuals meet the normality criteria with mean 0.
6. Your model is not validated against actual data. Is there any reason for this?
7. What is the significance of your findings in actual planning practice to improve traffic safety?
Author Response
1. Equations (2-1) and (2-2) refer to UTAM. Is UTAM different than TMAU? If they are the same,
please be consistent - use only either UTAM or TMAU, not both. If they are different, then please
define UTAM.
Observation accepted and adjusted. Equations (2-1) and (2-2) refer to TMAU, this was a typing
error.
2. Please explain all variables in Table 2.1. For example, what constitutes the mean of Land Use Factors
(LV)? When mean LV = 0.55, how is it computed, i.e., mean of what? What is the unit of LV?
Observation accepted and adjusted. Variables in table 2.1. were explained detailing the units.
In the case of the latent variable (LV) Land uses and socioeconomic stratification since it is a
vector that represents the explained weighted variance of these categorical variables, the statistics
appear in table 2.1. represent the measures of central tendency and dispersion for this vector.
It is a percentage measure. In lines 300 to 307, the explanation of how the latent variable was
constructed is adjusted.
3. How is W (spatial weighting matrix) computed? What is the equation for computing W? What
does W represent? Is there any learning algorithm used to compute W?
As I mentioned in section 3.2. the matrix W is a non-negative N×N matrix that describes the spatial configuration or arrangement between spatial units (TMAU). This was defined by two criteria cited in section 3.1: based on physical contiguity of order one (rock) and based on neighborhood graphs (relative neighbors' graph), the equations of these criteria can be consulted in the citations [31] and [32] and for the calculation of W, the spatial dependence “spdep” package of R was used [33].
4. Why GNS and SDEM selected as your spatial regression models? What are your justifications?
Are they better than other spatial regression models?
Observation accepted and adjusted. Lines 222 and 223 justify the reasons why the GNS and SDEM models were selected to estimate the TAI and TADI. To clarify, different spatial linear regression models were tested to estimate TAI and TADI, evaluating the fulfillment of the traditional Gauss-Markov assumptions and the adjustment and performance of these using MAE and RSME. In the case of the TAI, the GNS model obtained better results compared to the others, while for the TADI the SDEM model was the only one in which all the traditional assumptions of Gauss-Markov were fulfilled. In addition, these allowed modeling the spatial dependence present in the variable of interest.
5. In Line 225, the CLRM model requires the residual e~N(0,sd2). Nowhere in the paper you test whether the residuals meet the normality criteria with mean 0.
The acronyms S-W are adjusted to Shapiro Wilk in table 4.2. As I mentioned in lines 310 and 311, the normality criterion in the residuals was evaluated using the Shapiro-Wilk test, which
posits the null hypothesis that the residuals come from a normal distribution. This was corroborated in table 4.2. (Shapiro – Wilk Residuals) where at a significance level of 0.05 the null
hypothesis is not rejected, concluding that the residuals come from a normal distribution.
6. Your model is not validated against actual data. Is there any reason for this?
The models were not validated with more current information because the entities in charge of monitoring and controlling road safety in the city openly make the data available to users with a time lag of approximately two years. Therefore, the data used in this study is based on the most recent version provided by official sources.
7. What is the significance of your findings in actual planning practice to improve traffic safety?
In the paragraph between lines 462 and 468, the importance of the findings obtained in this study was described as improving road safety in an urban environment such as the city of Bogotá.
Reviewer 2 Report
I recommend expanding the analysis in another study in the future.
What is the main question addressed by the research? Is it relevant and interesting?
It is relevant and interesting.
How original is the topic? What does it add to the subject area compared with other published material?
It is a current topic
Is the paper well written? Is the text clear and easy to read?
very well structured.
Author Response
1. What does it add to the subject area compared with other published material?
The impact of socioeconomic, land use and mobility variables on the frequency of traffic accidents and deaths caused by them in an urban environment was evaluated using spatial linear regression and support vector models. The spatial weighting matrix was included as an input parameter in the support vector regression models to evaluate their performance against those that ignore the spatial dependence
structure of the vehicular accident phenomenon. The results showed that the support vector regression models that consider the spatial dependence obtained better results even than the traditional spatial econometric models.